

Ice sheet model simulations reveal polythermal ice conditions existed across the NE USA during
the Last Glacial Maximum
Joshua Cuzzone[1], Aaron Barth[2], Kelsey Barker[2], Mathieu Morlighem[3]
[1]Joint Institute for Regional Earth System Science and Engineering, University of California, Los
Angeles, USA
[2]Department of Geology, Rowan University, Glassboro, USA
[3]Department of Earth Sciences, Dartmouth College, Hanover, USA
*Correspondence to*: Joshua K. Cuzzone (jcuzzone@ucla.edu)
**Abstract**
Geologic evidence of the Laurentide Ice Sheet (LIS) provides abundant constraints on the areal
extent of the ice sheet during the Last Glacial Maximum (LGM). Direct observations of LGM LIS
thickness are non-existent, however, with most geologic data across high elevation summits in the
Northeastern United States (NE USA) often showing signs of inheritance, indicative of weakly
erosive ice flow and the presence of cold-based ice. While warm-based ice and erosive conditions
likely existed on the flanks of these summits and throughout neighboring valleys, summit
inheritance issues have hampered efforts to constrain the timing of the emergence of ice-free
conditions at high elevation summits. These geomorphic reconstructions indicate that a complex
erosional and thermal regime likely existed across the southeasternmost extent of the LIS
sometime during the LGM, although this has not been confirmed by ice sheet models. Instead,
current ice sheet models simulate warm-based ice conditions across this region, with disagreement
likely arising from the use of low resolution meshes (e.g., >20 km) which are unable to resolve the
high bedrock relief across this region that strongly influenced overall ice flow and the complex
LIS thermal state. Here we use a newer generation ice sheet model, the Ice-sheet and Sea-level
System Model (ISSM), to simulate the LGM conditions of the LIS across the NE USA and at 3
localities with high bedrock relief (Adirondack Mountains, White Mountains, and Mount
Katahdin), with results confirming the existence of a complex thermal regime as interpreted by the
geologic data. The model uses higher-order physics, a small ensemble of LGM climate boundary
conditions, and a high-resolution horizontal mesh that resolves bedrock features down to 30 meters
to reconstruct LGM ice flow, ice thickness, and thermal conditions. These results indicate that
across the NE USA, polythermal conditions existed during the LGM. While the majority of this
domain is simulated to be warm-based, cold-based ice persists where ice velocities are slow (<15
m/yr) particularly across regional ice divides (e.g., Adirondacks). Additionally, sharp thermal
boundaries are simulated where cold-based ice across high elevation summits (White Mountains
and Mount Katahdin) flank warm-based ice in adjacent valleys. Because geologic data is
geographically limited, these high-resolution simulations can help fill gaps in our understanding
of the geographical distribution of the polythermal ice during the LGM. We find that the elevation
of this simulated thermal boundary ranges between 800-1500 meters, largely supporting geologic
interpretations that polythermal ice conditions existed across NE USA during the LGM, however
this boundary varies geographically. In general, we show that a model with finer spatial resolution
and higher order physics is able to simulate the polythermal conditions captured in the geologic
data, with model output being of potential utility for site selection in future geologic studies and
geomorphic interpretation of landscape evolution.




## 1. Introduction

During the Last Glacial Maximum (26.5 to 19.0 ka; Clark et al., 2009), global temperatures cooled by ~6.1ºC (Tierney et al., 2020) leading to the growth of expansive ice sheets and the lowering of global sea level by ~130 m (Clark & Mix, 2002). As part of the North American Ice Sheet complex (NAIS), the Laurentide Ice Sheet (LIS) is estimated to have contained 75-85 m global sea-level equivalent (SLE; Clark & Mix, 2002) thus representing a climatically important component of the cryosphere. Extending southwards from its source region in northern Canada, the LIS covered most of the northeastern United States (NE USA; Fig. 1) with its terminal position located along Martha's Vineyard, Massachusetts (MA), Long Island, New York (NY), and into northern New Jersey and Pennsylvania to the west (Dalton et al., 2020). The retreat of the LIS during the last deglaciation is constrained through numerous geochronologic studies including: varve chronologies (Ridge et al., 2012), basal radiocarbon dates (Fig., 1d; Dyke et al., 2004; Dalton et al., 2020), and terrestrial in-situ cosmogenic nuclide surface exposure ages of moraines (Balco and Schaefer, 2006; Bromley et al., 2015; Ullman et al., 2016; Hall et al., 2017; Bromley et al., 2020; Balter-Kennedy et al., 2024). While these geologic archives constrain ice margin retreat well, the vertical thinning history and ultimately the volumetric change of the LIS during the last deglaciation in this region remains poorly known.

Fortunately, because of the high vertical relief across the NE USA, studies have addressed the vertical thinning history of the LIS in this region by dating glacial features along vertical transects (herein referred to as dipstick studies; Bierman et al., 2015; Koester et al., 2017; Barth et al., 2019; Corbett et al., 2019; Koester et al., 2020; Halsted et al., 2023). These studies indicate that rapid vertical ice sheet thinning occurred coincident with ice margin retreat during the last deglaciation, and predominantly during the Bølling-Allerød warm period. Through surface exposure dating of bedrock features and glacial erratics, these dipstick studies commonly find the presence of inherited nuclides across high elevation sites (>1200 m; Halsted et al., 2023), making geologic interpretations of the onset of vertical ice thinning at these locations difficult. Consequently, these data suggest that the high elevation regions of the NE USA were likely covered by cold-based ice characterized by the absence of subglacial water and ultimately much reduced subglacial erosion, with warm-based ice and erosive conditions flanking the valleys of these high elevation regions (Halsted et al., 2023). While these geologic interpretations support the existence of polythermal ice conditions across the NE USA, it is not well known how this subglacial regime varied spatially and whether the existence of this boundary occurred along a geographically consistent elevation. This has implications for interpreting geologic data of past ice sheet retreat or thinning particularly at high elevations, as well as erosional processes that may have operated during glaciation across the NE USA, as erosional patterns are closely related to the ice sheet thermal regime (Lai and Anders et al., 2021). Ultimately, where these geologic archives are limited in their spatial coverage, ice sheet models can be used to simulate broader characteristics of this thermal regime.

Ice sheet models have been important tools to study LIS conditions during the LGM and assess the drivers of deglacial change (Sugden, 1977; Marshall et al., 2000; Hooke and Fastook, 2007; Tarasov and Peltier, 2007; Gregoire et al., 2012; Moreno et al., 2023). Not only can ice sheet models aid in the interpretation of geologic proxies of past ice sheet behavior, but they can provide outputs that enable a more informed choice when considering field locations for sampling. For example, a recent assessment of the basal thermal state of the Greenland Ice Sheet (MacGregor et al., 2016; 2022), which in part relied upon output from newer generation ice sheet models, was used to inform field campaigns aimed at sampling subglacial bedrock in portions of the Greenland



Ice Sheet that were estimated to be cold-based with low erosion (Briner et al., 2022).  In regards
to the LIS, prior ice sheet modeling efforts were useful in identifying a broad picture of the thermal
behavior of the LIS, which indicated that at the LGM roughly 20-50% of the LIS was warm based
(Marshall and Clark, 2002; Tarasov and Peltier, 2007), including the NE USA.  Some aspects of
these simulations (Tarasov and Peltier, 2007) do agree well with broad scale geomorphic indicators
of basal conditions (Klemen and Hattestrand, 1999; Briner et al., 2006; Klemen and Glasser, 2007;
Briner et al., 2014), which indicate that the LIS exhibited a varying subglacial thermal regime,
with frozen bed patches interspersed particularly along ice divides and in some cases residing
along sharp boundaries with warm-based ice streams or outlet glaciers.  However, due to the low
spatial resolution of existing models (>20 km), the high topographic relief across the NE USA is
poorly resolved and therefore the sharp thermal boundary between cold and warm-based ice
identified in the geologic archives is not captured. This is likely due to the more coarsely resolved
models' inability to capture advective and diffusive processes at these small spatial scales.
Likewise, as the high relief of this region served as a control and impediment to ice flow during
the LGM, the impacts of ice flow on deformational and frictional heating is more poorly captured
in lower resolution models.  This can be improved by downscaling ice sheet models to higher
resolution, which has shown promise (Staiger et al., 2005) in interpreting how polythermal ice
conditions may have influenced regional glacial geomorphology.
To address this shortcoming, we use a high-resolution ice sheet model to simulate the
thermal regime of the LIS across the NE USA during the LGM.  Our model experiments are
designed to test whether the presence of this sharp thermal regime is simulated and to assess the
spatial and elevational characteristics of the basal thermal regime at regional and local scales (i.e.,
mountain range) where geologic evidence suggests the existence of polythermal conditions.
Through this work, we aim to support current geologic interpretations, while also filling gaps in
our understanding of this complex thermal regime where geologic constraints are limited.

### *2. Methods*

The NE USA, comprising of the states listed in Figure 1, is marked by high topographic
relief that spans an elevational range from sea-level to >1500 meters. In order to capture these
large gradients in bedrock topography and thermal boundaries within the ice sheet, we employ a
nested modelling approach (Briner et al., 2020; Cuzzone et al., 2022), whereby a more coarsely
resolved and larger model provides boundary conditions necessary for downscaling over regional
and local scale domains that are more finely resolved (see Figure S1).

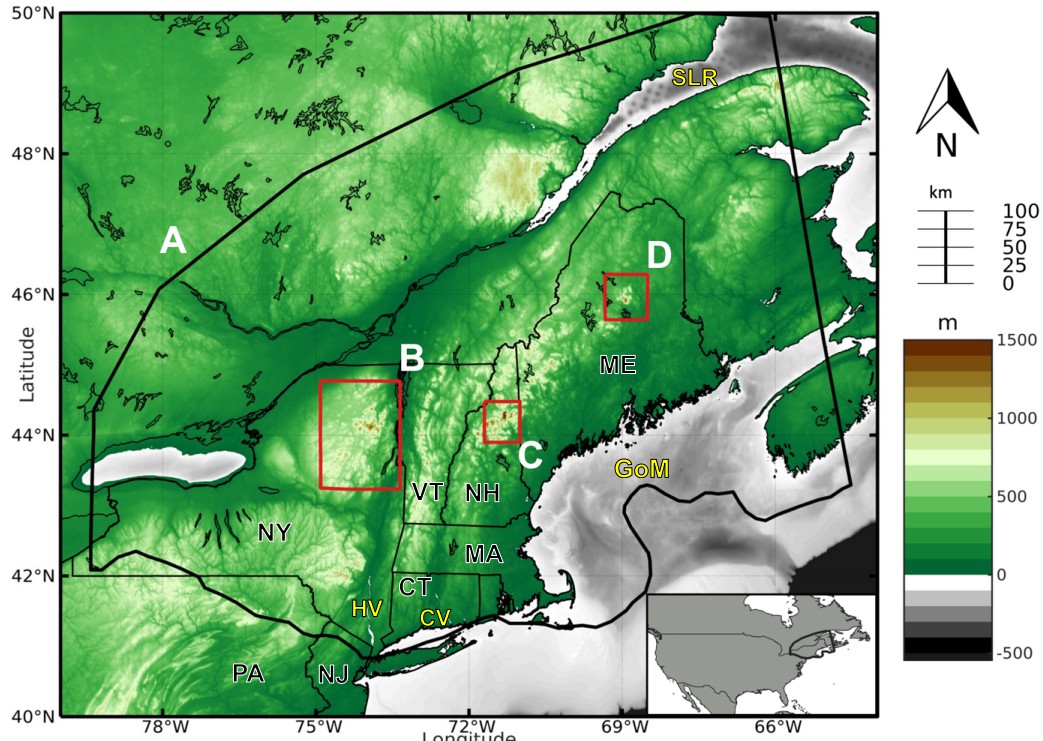

Figure 1. Bedrock topography (meters) and model domains for the regional NE USA ice sheet model (shown as black outline; A) and the local scale models shown in the red outlines across the Adirondack Mountains (B), White Mountains (C) and Mount Katahdin (D). The southern boundary of the NE USA model (A) follows the reconstructed LGM ice extent at the LGM from Dalton et al. (2020). States are abbreviated as: PA: Pennsylvania, NJ: New Jersey, CT: Connecticut, NY: New York, MA: Massachusetts, VT: Vermont, NH: New Hampshire, ME: Maine HV: Other locations described in the text are abbreviated as: Hudson Valley, CV: Connecticut Valley, GoM: Gulf of Maine, SLR: Saint Lawrence River Valley.

*2.1.1 Ice Sheet Model*
We use the Ice-sheet and Sea-level System Model (ISSM; Larour et al., 2012), a higher-
order thermomechanical finite-element ice sheet model to simulate the LIS LGM conditions across
the NE USA. Because of the high topographic relief across this region and associated impact on
ice flow, we use a higher-order approximation to solve the momentum balance equations (Dias dos
Santos et al., 2022). This ice flow approximation is a depth-integrated formulation of the higher-
order approximation of Blatter (1995) and Pattyn (2003), which allows for an improved
representation of ice flow compared with more traditional approaches in paleo-ice flow modelling
(e.g., shallow ice approximation or hybrid approaches).
An enthalpy formulation that simulates both temperate and cold-based ice (Aschwanden et
al., 2012; Serrousi et al., 2013; Rückamp et al., 2020) is used to capture the thermal state of the ice
sheet. Our model contains four vertical layers and uses quadratic finite elements (P1 × P2) along



the z axis for the vertical interpolation following Cuzzone et al. (2018) in order to better capture
sharp thermal gradients near the base and simulate the vertical distribution of temperature within
the ice. This methodology has been successfully applied to simulate the transient behavior of the
Greenland Ice Sheet across geologic timescales and the contemporary period (Briner et al., 2020;
Smith-Johnsen et al., 2020; Cuzzone et al., 2022). The ice rheology follows Glen's flow law with
the ice viscosity being dependent on the simulated ice temperature following rate factors in Cuffey
and Paterson (2010). Surface temperature (see 2.1.3) and geothermal heat flux (Shapiro and
Ritzwoller, 2004) are imposed as boundary conditions in the thermal model. We use a linear
friction law (Budd et al., 1979), where the basal drag ($\tau_b$) is represented as:
$$\tau_b = -\alpha^2 N u_b \tag{1}$$
where $\alpha$ represents the spatially varying friction coefficient, $N$ represents the effective pressure,
and $u_b$ is the basal velocity. Here $N = g(\rho_i H + \rho_w Z_b)$, where $g$ is gravity, $H$ is ice thickness, $\rho_i$ is
the density of ice, $\rho_w$ is the density of water, and $z_b$ is bedrock elevation following Cuffey and
Paterson (2010). $N$ evolves as the ice sheet thickness changes. The spatially varying friction
coefficient ($\alpha$) is constructed as a function of bedrock elevation above sea level (Åkesson et al.
2018; Cuzzone et al., 2024):
$$\alpha = 200 \times \frac{\min[\max(0, z_b + 600), z_b]}{\max(z_b)} \tag{2}$$
where $z_b$ is the height of the bedrock with respect to sea level. Using this parameterization, basal
friction is larger across high topographic relief and lower across valleys and areas below sea level,
which is consistent with what is found today for the Greenland Ice Sheet. The thermomechanical
coupling captures the impact of ice deformation and frictional heating on ice temperature, which
in turns affect the ice rheology through the temperature dependent rate-factor. In our approach, the
friction coefficient is independent of the thermal state but has been explored recently in modeling
LIS LGM conditions (Moreno et al., 2023). Although it is found to have a measurable impact the
overall simulated LGM ice volume, basal temperatures, and ice stream extent particularly for
Hudson Bay, these variations are small when compared with uncoupled simulations particularly
across the NE USA.
The motion of the ice front is tracked using the level-set method described in Bondzio et
al. (2016) and calving is simulated where the LIS interacts with the ocean based on the Von Mises
stress criterion (Morlighem et al., 2016). This approach approximates the calving rate as a function
of the tensile stresses simulated with the ice, with ice front retreat occurring when the von Mises
tensile strength exceeds user defined stress thresholds set for tidewater (1 MPa) and floating ice
(200 kPa). These values are consistent with other contemporary and paleo ice sheet modeling
studies (Bondzio et al., 2016; Morlighem et al., 2016; Choi et al., 2021; Cuzzone et al., 2024).
Glacial isostatic adjustment (GIA) is not simulated transiently, but is instead prescribed by
using a reconstruction of relative sea level from a global GIA model of the last glacial cycle (Caron
et al., 2018). Bedrock vertical motion, eustatic sea-level, and geoid change at the LGM are applied
to our model to account for GIA (Briner et al., 2020; Cuzzone et al., 2024).
*2.1.2 Model Domains*
To constrain the model boundary conditions necessary for simulating the LIS conditions
across the NE USA at both a regional (Figure 1; Domain A) and local scales across the Adirondack



mountains, White Mountains, and Mount Katahdin (Figure 1; Domain B-D), we first simulate the LGM ice conditions across the LIS (Figure S1). Our model domain follows the LGM ice extent reconstructed from Dalton et al. (2020) but does not include the Cordilleran Ice Sheet and the connection between Ellesmere Island and the Greenland Ice Sheet. The LIS model is built on a structured mesh with a spatial resolution of 20 km. Bedrock topography is initialized from the General Bathymetric Chart of the Oceans (GEBCO; GEBCO Bathymetric Compilation Group, 2021), which relies on the 15 arcsecond (450-meter resolution) Surface Radar Topography Mission data (SRTM15_plus; Tozer et al., 2019) for the terrain model. The regional ice sheet model domain (Figure 1, Domain A; Figure S1, Domain 2) covers the NE USA and extends into portions of southern and maritime Quebec. Across this domain anisotropic mesh adaption is used to construct a non-uniform mesh that varies based upon gradients in bedrock topography from GEBCO. The spatial resolution varies from 5 km in areas of low topographic relief to 500 meters in areas of high topographic relief. To gain a more detailed reconstruction of the LGM conditions across the NE USA that are more consistent with the spatial scales associated with the geomorphic data (Halsted et al., 2023), we downscale our results to three local areas within the NE USA (Section 2.2; step 3): the Adirondack Mountains, New York; the White Mountains, New Hampshire; and Mount Katahdin, Maine. The local scale models (Figure 1, Domains B-D; Figure S1, Domain 3) also rely on anisotropic mesh adaptation with spatial resolution varying from 400 to 30 meters based upon topography from the 1 arcsecond (~30-meter resolution) SRTM product (Farr et al., 2007). The three locations were chosen based on their geomorphic qualities, as each represents the highest peak within their respective state, and has pre-existing geologic data on glaciation (e.g., cosmogenic surface exposure ages; Bierman et al., 2015; Barth et al., 2019). At these spatial scales, the model is able to resolve the high topographic relief found between the mountain peaks and neighboring valley floors.

2.1.3 Climate Forcing and Surface Mass Balance scheme

Output of monthly mean LGM temperature and precipitation from 5 different climate models is used as surface boundary conditions and to estimate the surface mass balance (SMB). The LGM climate from the TraCE-21ka transient simulation of the last deglaciation is used (Liu et al., 2009; He et al., 2013), as well as output from 4 models participating in the Paleoclimate Modelling Intercomparison Project 4 (PMIP4; Kageyama et al., 2021). The simulated climate from these models differs both with respect to the magnitude and spatial distribution of glacial temperature and precipitation change relative to the preindustrial climate (Figure S2). By using a diversity of surface climate forcings rather than the multi-model mean, we aim to construct a small ensemble of results for the simulated LGM conditions across our model domains as the representative climate model spread can impact the simulated ice sheet geometry, ice flow, and thermal characteristics (Lai and Anders, 2021).

| Model | Spatial Resolution | Reference |
|-------|-------------------|-----------|
| CCSM4 (TraCE-21ka) | 3.75° x 3.75° | Liu et al., 2009<br>He et al., 2013 |
| MPI-ESM1.2 (MPI) | 1.8° x 1.8° | Mauritsen et al., 2019 |
| MIROC-ES2L (MIROC) | 2.8° x 2.8° | Ohgaito et al., 2021<br>Hajima et al., 2020 |
| IPSLCM5A2 (IPSL) | 3.8° x 1.9° | Sepulchre et al. (2020) |
| AWIESM2 (AWI) | 1.8° x 1.8° | Sidorenko et al. (2019) |





Table 1. List of climate models used for the LGM surface climate forcing and the corresponding spatial resolution.

We use a positive degree day model (Tarasov and Peltier, 1999; Le Morzadec et al., 2015; Cuzzone et al., 2019; Briner et al., 2020) to calculate the SMB. Our degree day factor for snow melt is 5 mm ºC$^{-1}$day$^{-1}$ and 9 mm ºC$^{-1}$day$^{-1}$ for bare ice melt, and we use a lapse rate of 5ºC/km to adjust the temperature of the climate forcings to surface elevation (Abe-Ouchi et al., 2007). The hourly temperatures are assumed to have a normal distribution, of standard deviation 3.5ºC around the monthly mean. An elevation-dependent desertification is included (Budd and Smith, 1981), which reduces precipitation by a factor of 2 for every kilometer change in ice sheet surface elevation and the model accounts for the formation of superimposed ice following Janssens and Huybrechts (2000). The degree day model requires inputs of monthly temperature and precipitation. We apply a commonly used modeling approach to scale a contemporary climatology of temperature and precipitation back to the LGM ('Anomaly Method'; Pollard et al., 2012; Seguinot et al., 2016; Golledge et al., 2017; Tigchlaar et al., 2019; Briner et al., 2020; Cuzzone et al., 2022). The monthly mean climatology of temperature and precipitation for the period 1979-2010 from the European Center for Medium-Range Weather Forecasts ERA5 reanalysis (Hersbach et al., 2020) is bilinearly interpolated onto our model mesh. Next, anomalies of the monthly mean temperature and precipitation fields from the climate models (Table 1), computed as the difference between the LGM and preindustrial control run, are added to the contemporary monthly mean climatology to produce the monthly temperature and precipitation fields at LGM.

*2.2 Experimental Setup*

Our downscaling approach is conducted over 3 steps:

*Step 1 (LIS models; Figure S1 and S3):* First, we construct a model of the LGM LIS using a 2d model with setup described above, and prescribe constant LGM climate (section 2.1.3). Following Moreno et al. (2023), we initialize our model with ice thicknesses of 1000 m north of 50°N, and allow the simulated LIS to reach equilibrium with respect to ice volume (~50,000 years). The resulting models (ice geometry and velocity) are used to initialize a 3d model that extrudes the 2d model to 4 vertical layers (P1xP2 vertical finite elements; section 2.1.1). The thermal regime is assumed to be in steady-state with the LGM climate, and we perform a thermomechanical steady state calculation with a fixed ice sheet geometry until the ice sheet velocities are consistent with ice temperature (i.e., convergence) following Seroussi et al. (2013). Lastly, we let the 3d LIS model relax an additional 20,000 years (i.e., until thermal equilibrium is reached) with the LGM climate as the model adjusts to the updated ice temperature and rheology.

*Step 2 (NE USA models; Figure 1)*: We construct our 3d NE USA ice models following the setup discussed in section 2.1 and initialize the model with ice geometry, temperature and rheology, and velocity from the resulting LGM 3d LIS model. Boundary conditions of temperature, ice velocity, and thickness from the LIS results are imposed as Dirichelt boundary conditions at the western, northern, and eastern boundaries following Cuzzone et al. (2022). The initial ice velocity and temperature are downscaled across this domain by performing a thermomechanical steady state calculation (Seroussi et al., 2013). The model is then allowed to relax with constant LGM climate for 20,000 years as the ice geometry, flow, and temperature adjust to the higher resolution grid.



*Step 3 (local models; Figure 1)*: Similar to *Step 2*, the local scale models are initialized with ice geometry, temperature and rheology, and flow with the results from the NE USA model, and boundary conditions (as in *Step 2*) are applied from the NE USA model to the local scale models at the North, South, East, and West boundaries. After running a thermomechanical steady state calculation, the model is allowed to relax for 20,000 years with constant LGM climate as the ice geometry, flow, and temperature adjust to the higher resolution grid.

Our approach above makes use of the thermomechanical steady state calculation to avoid high computational expense in relaxing the 3d models for a sufficiently long time (e.g. >100,000 yr) until thermal equilibrium is reached. For this study, we focus the discussion of our results on the simulated LGM state of the NE USA (step 2) models and local models (step 3), but provide Figure S3 to illustrate the simulated LGM state for the LIS (step 1).

## 3. Results

### 3.1 Northeast USA

We simulate a range of possible LGM states given different climatologies, and do not specifically tune our models to match reconstructions of ice extent or flowlines. The southern boundary of our model domain is constrained to the maximum reconstructed LGM ice extent from Dalton et al. (2020). In total we have 5 different simulations of the LIS across the NE USA during the LGM (Figure S4), driven by the independent climate forcings (Table 1). The depth integrated ice velocity and ice thickness for the ensemble mean (n=5) of those experiments is shown in Figure 2. Individually, most of the experiments simulate an LGM ice margin that reaches the reconstructed terminal LGM ice extent (Figure 2B & S4) from Dalton et al. (2020), although some simulate reduced ice extent particularly along the southern and eastern boundaries (Figure S4). Thinner ice (<500 m) is found along the ice margin and marine terminating portions of the Atlantic Ocean and Gulf of Maine, and thickens to upwards of 3500 m over the Northwest portion of the model domain. Across the Northwest region, ice velocities are slow (<20 m/yr), consistent with a northward trending ice divide simulated through Quebec (see Figure S3). Additionally, a regional ice divide is simulated across the Adirondack mountains (Figure 2A), where ice velocities are <15m/yr and ice flow vectors indicate diverging ice flow to the southwest and southeast. This regional ice divide is simulated for all experiments (Figure S4), with variation in the magnitude of the ice velocities (~1-25 m/yr). Faster ice flow (>50 m/yr) is found along the ice margin and through areas of topographic troughs. Horizontal ice flow is fastest in marine terminating portions such as the Gulf of Maine (GoM; Figure 1), the St. Lawrence River (SLR; Figure 1), and throughout lower terrain such as the Hudson and Connecticut river valley (Figure 2A; HC, CV Figure 1), with speeds approaching and exceeding 300 m/yr.





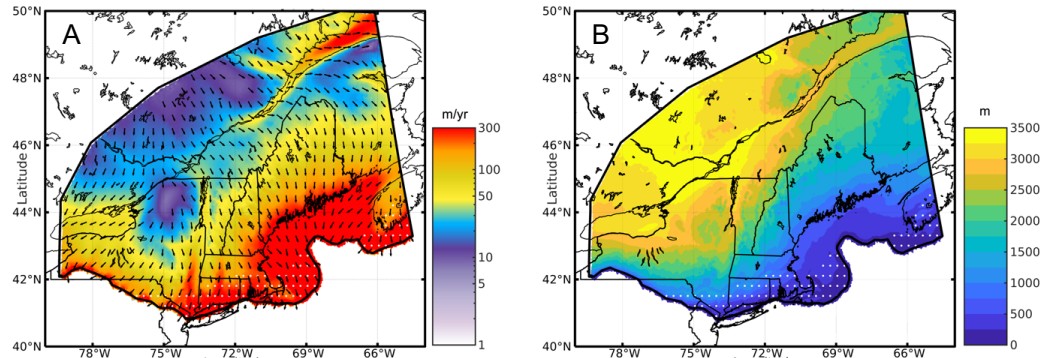

Figure 2. The modeled mean (n = 5; Table 1), simulated LGM depth-integrated ice velocity (A; m/yr) and simulated LGM ice thickness (B; m) across the NE USA. Vectors (A) illustrate the simulated direction of ice flow, and colors denote the magnitude of ice velocity. White stippling indicates where areas where some individual models (see Figure S4) simulate no LGM ice cover.


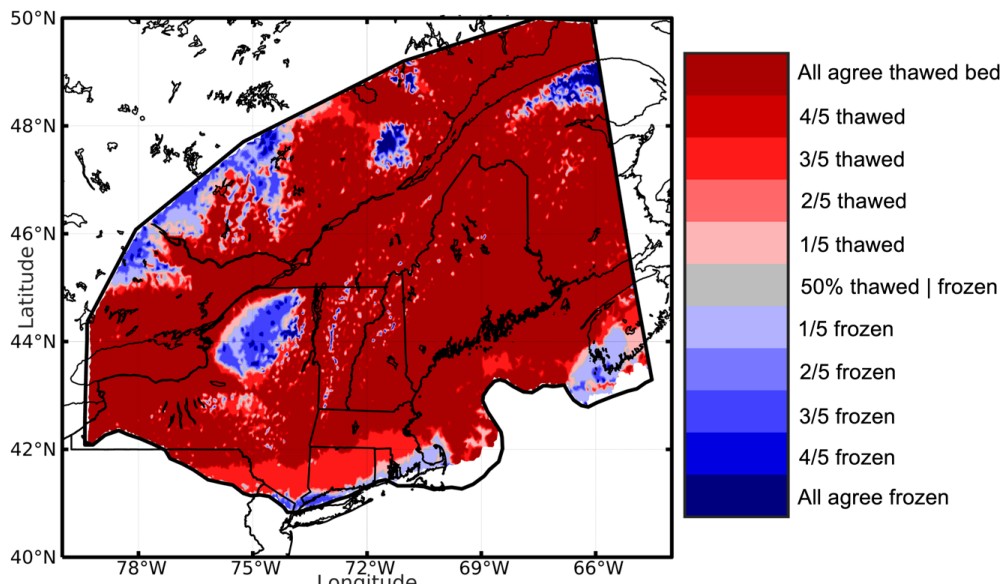

Figure 3. The agreement in the simulated LGM basal thermal state (n = 5) for the experiments with varying climate forcing. We assume a thawed bed is present when the pressure corrected $T_{bed} \geq -1°C$ and frozen bed is present when $T_{bed} < -1°C$ following MacGregor et al., 2022.

310   For each experiment, we calculate the simulated basal temperature, adjusted for pressure
311 melting (i.e., pressure corrected temperature). Following MacGregor et al. (2022), we consider a
312 thawed bed to be simulated where the pressure corrected $T_{bed} \geq -1°C$ and frozen where $T_{bed} < -1°C$.



The magnitude of the simulated thawed and frozen bed conditions varies across our model domain
(Figure S4) but follow a spatial pattern that can be summarized in Figure 3, where we show the
inter-model agreement between experiments for thawed and frozen bed conditions.  We find that
approximately 70% of the model domain is thawed and only 2% of the model domain has frozen
bed conditions (i.e., area of domain where all experiments agree), indicating that warm-based ice
conditions prevailed across this portion of the LIS. Warm-based areas are coincident with areas of
fast ice flow (Figure 2A), where sliding dominates.  Frozen bed conditions are generally simulated
in areas of reduced ice flow (<15 m/yr; Figure 2A), particularly along the ice divide that exists in
the northwest portion of the model domain, south of the St. Lawrence River, and across the
Adirondack mountains.  Additionally, the regional model simulates the presence of cold-based ice
scattered across areas of high bedrock elevation (see Figure 1 for topographic map).  In general,
each experiment using the different climate surface forcings simulate a similar spatial pattern of
warm-based and cold-based ice (Figure S4).  Where each experiment disagrees is with respect to
the magnitude of basal cooling, with some experiments simulating both colder basal temperatures
and a wider swath of frozen bed conditions primarily across the Adirondack Mountains (Figure
S4; TraCE-21ka, MPI, and IPSL).  This is likely related to the differences in the simulated LGM
surface climate between each climate model.  The experiments which simulate a larger swath of
frozen bed conditions across the Adirondack Mountains (Figure S4; TraCE-21ka, MPI, and IPSL)
have a higher magnitude of simulated LGM surface cooling and higher SMB (Figure S5; TraCE-
21ka, MPI, and IPSL), versus the experiments which simulate less extensive frozen bed conditions
across the Adirondack Mountains (Figure S4 and S5; MIROC and AWI).
*3.2 Local Models*
*3.2.1 Adirondacks*
The simulated mean (n=5) LGM conditions for the Adirondacks are presented in Figure 4.
High topographic relief that exceeds 1000 m is characteristic of the Adirondack Mountains, with
43 mountain peaks exceeding 1200 m in elevation, and maximum elevations reaching 1600 m
(Figure 4; HPA: High Peaks area).  Simulated LGM ice thickness reaches between 1600 m to 2000
m across the highest peaks, increases throughout the lower valleys in excess of 2000 m, and reaches
upwards of 3000 m across the lower elevations in the Northwest and Northeast portion of the
domain (Figure 4C).  Consistent with the ice divide simulated in the regional model (Figure 2),
slow ice velocities (<10 m/yr to 25 m/yr) dominate across the Adirondacks (Figure 4D).  Ice flow
generally trends in a south-southeastward direction, with ice velocities increasing to > 60 m/yr
across lower elevation valleys.  Approximately 58% of the model domain is thawed and 4% of the
model domain has frozen bed conditions (i.e., area of domain where all experiments agree),
indicating that warm-based ice conditions prevailed across this portion of the LIS (Figure 4E).
Generally, frozen bed conditions are confined to high elevation peaks (Figure 4B), where low ice
velocities and thinner ice exist, making these high elevation regions more susceptible to vertical
advection of cold surface temperature (LGM temperature range from climate models: *-24°C to -*
*18°C*).  However, we also find that across lower elevation regions, particularly in the Northwest
portion of the domain, frozen bed conditions are simulated.  Here ice is thick (upwards of 3000 m;
Figure 4A) but is characterized by very slow ice flow (<10 m/yr).  Without limited frictional and
deformational heating, this area was likely influenced by the vertical advection of cold surface
climate despite thick ice.  Across this domain the thermal boundary separating frozen and thawed
bed (Figure 4E) is 882 m.  However, if we just consider the High Peaks area (HPA; Figure 4A),
that thermal boundary resides at 1180 m.

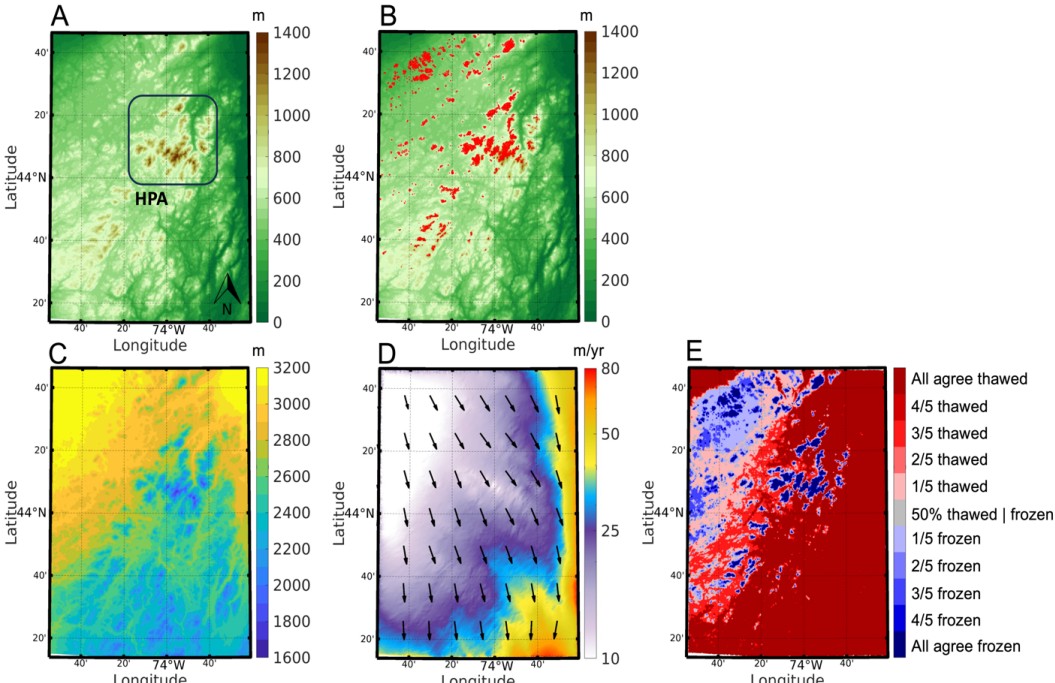

Figure 4. A) Bedrock topography for the Adirondack Mountains (m). Box highlights the High Peaks area. See Figure 1 for geographical context within the NE USA. B) Bedrock topography for the Adirondack Mountains with an overlay of areas where the models agree (n=5) on simulated frozen bed conditions. C) The model mean ice thickness (m). D) The model mean depth average ice velocity (m/yr). Note, vectors denote ice flow direction and not the magnitude of velocity. E) Modeled agreement for simulated frozen and thawed bed conditions.


*3.2.2 White Mountains*

To the east of the Adirondack Mountains, the White Mountains comprise a series of
mountain ranges intersected by deeply incised valleys, with the highest peak Mount Washington
reaching 1912 m (Figure 5A). The simulated mean LGM conditions for the White Mountains are
presented in Figure 5. Ice thickness ranges between 600 m to 1200 m across high elevations peaks
(Figure 5C) and thickens to 2000 m in the deeply incised valleys. Across the lower elevations in
the Northwest portion of the model domain (Figure 5A), ice thickness reaches upwards of 2500
m. Ice flows southeastward across this region, with simulated ice velocities being higher than is
found across the Adirondack Mountains. Ice velocities are lowest in the Northwest portion of the
domain and across Mount Washington, where the southeastward ice flow is impeded by the high
elevation bedrock (35 – 60 m/yr; Figure 5D), and increases up to 150 m/yr across the lower
elevations in the southeast region. When considering the basal thermal regime, we find that 95%
of the model domain has thawed bed conditions, with only 0.5% of the domain having frozen bed
conditions. Frozen bed conditions are limited to high elevation sites, particularly across the
Presidential Range where Mt. Washington is located and Mount Lafayette and Little Haystack in
the Franconia Range, with a mean thermal boundary between frozen and thawed bed conditions
residing at 1530 m.

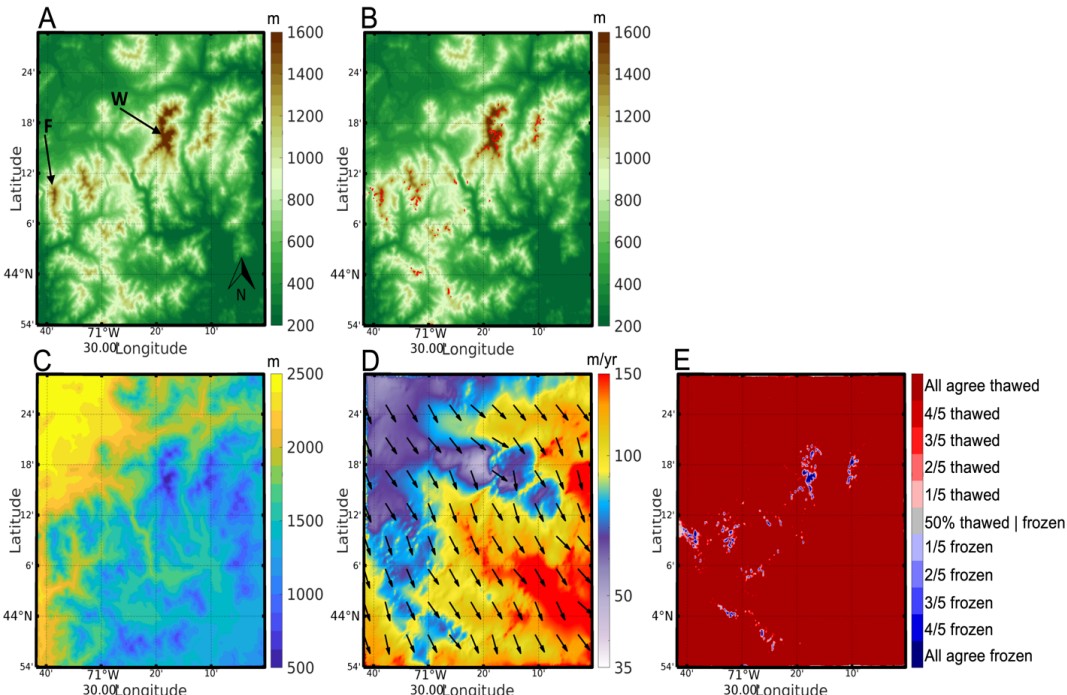

Figure 5. A) Bedrock topography for the White Mountains (m). Highlighted are Mt. Washington (W) and Mount Lafayette and Little Haystack Mountain, labeled (F) for Franconia Range. See Figure 1 for geographical context within the NE USA. B) Bedrock topography for the White Mountains with an overlay of areas where the models agree (n=5) on simulated frozen bed conditions. C) The model mean ice thickness (m). D) The model mean depth average ice velocity (m/yr). Note, vectors denote ice flow direction and not the magnitude of velocity. E) Modeled agreement for simulated frozen and thawed bed conditions.


*3.2.3 Mount Katahdin*
Mount Katahdin reaches 1606 m with upwards of 1300 m of relief above the surrounding
valleys (Figure 6A). The simulated mean LGM conditions for Mount Katahdin are presented in
Figure 6. Across the summit of Mount Katahdin, ice thicknesses reach between 500-700 m, and
thickens to 1500 - 2000 m around the flanks of the mountain (Figure 6C) and the surrounding
lowlands. The ice generally flows south-southeastward (Figure 6D), with ice flow diverging and
slowing down to 25 m/yr upstream of Mountain Katahdin, before reaching a minimum of 10-15
m/yr at the summit. Ice flow converges on the downstream side of Mount Katahdin and reaches
50-100 m/yr across the lower elevations to the south and east. The simulated basal thermal regime
indicates that approximately 97% of the domain is thawed (Figure 6E), with only a few locations
across the summit of Mount Katahdin having frozen bed conditions, representing ~0.5 % of the
model domain. The elevational boundary separating frozen and thawed bed conditions across this
domain is simulated to be at 1318 m.






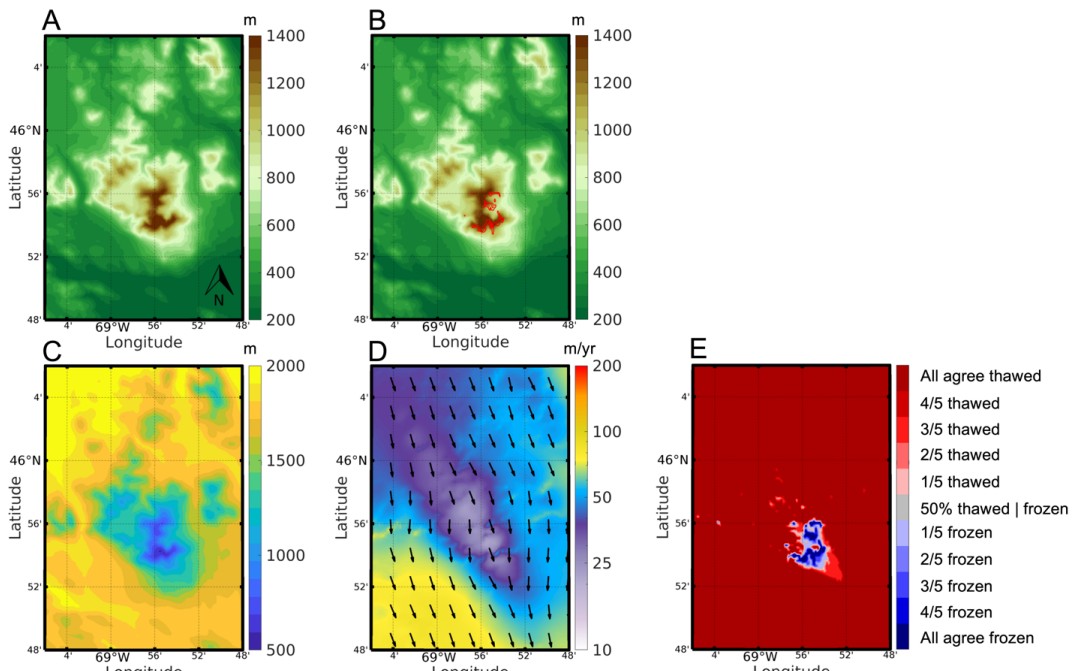

Figure 6. A) Bedrock topography for Mount Katahdin (m). See Figure 1 for geographical context within the NE USA. B) Bedrock topography for Mount Katahdin with an overlay of areas where the models agree (n=5) on simulated frozen bed conditions. C) The model mean ice thickness (m). D) The model mean depth average ice velocity (m/yr). Note, vectors denote ice flow direction and not the magnitude of velocity. E) Modeled agreement for simulated frozen and thawed bed conditions.


## 4. Discussion
Across the broader LIS, geomorphic evidence supports the existence of frozen bed
conditions interspersed amongst regions of warm-based ice (Klemen and Hattestrand, 1999;
Klemen and Glasser, 1997; Marquette et al., 2004) and along southern sectors of the LIS, where
ice cover was thin (Colgan et al., 2002). More regional indicators of polythermal conditions with
sharp contacts of warm-based ice in fast flowing outlet glaciers and cold-based ice on slow moving
ice on uplands have been found across Arctic Canada and Baffin Island (Davis et al., 1999; Davis
et al., 2006; Briner et al., 2006; 2014). Prior to direct evidence from surface exposure dating
(Bierman et al., 2015), it was unknown if cold-based ice may have existed across the NE USA.
Ice sheet models have been used to infer the basal thermal conditions of the LIS, with implications
for better understanding large-scale and regional ice flow and controls on ice mass evolution and
regional geomorphology (Sugden et al., 1977; Marshall and Clark, 2002; Tarasov and Peltier,
2007; Moreno-Parada et al., 2023). However, while these models provide possible scenarios for
LGM and the deglacial evolution of the LIS thermal state, the coarse spatial resolution of existing
models limits the ability to capture sharp thermal gradients that may have existed in high relief





terrain, with models simulating large scale warm-based conditions during the LGM across the NE USA (Tarasov and Peltier, 2007). Therefore, given that geologic and geomorphic indicators of LIS thermal conditions show that cold-based ice existed across areas of high relief in the NE USA (Goldthwait, 1940; Davis, 1989; Halsted et al., 2023), we relied upon a downscaling procedure to ensure that the underlying bedrock relief and resulting stress balance was well resolved. Our results agree with geologic interpretations (Davis, 1989; Bierman et al., 2015; Halsted et al., 2023) that suggest the existence of polythermal ice across the NE USA during the LGM. While the majority of this region was exposed to warm-based ice conditions (70%; Figure 3), frozen bed conditions are simulated across areas of low ice velocity (i.e. ice divides) and high elevations (2% of area in Figure 3). It is worth noting that while the focus of this study was on the NE USA, areas of frozen bed ice conditions are simulated for portions Maritime Canada (Figure 3), which agree reasonably well with geologic interpretations from this region (Olejczyk and Gray, 2007).

Because a majority of the geologic data constraining the thermal regime of the LIS across this region is found in areas of high relief, we focused our modeling on three specific locations. Regionally, geologic interpretations posit that the Adirondack Mountains may have acted as an impediment to the south-southeast flow of the LIS during glacial expansion and the LGM (Franzi et al., 2016). Our simulations suggest that ice flow across this region was slow (<15 m/yr; Figure 2A, 4A) in response to a divergence of ice flow around the mountainous terrain. Consequently, a regional ice divide and frozen bed conditions are simulated across portions of this region where ice velocities are <10 m/yr and in areas where the bedrock relief is high. This is further supported when looking at the individual model experiments. Our experiments relied on a small ensemble of simulated surface climate at the LGM from climate model experiments, each simulating a varying magnitude of LGM cooling and precipitation change. Those experiments using colder LGM boundary conditions and which simulated higher SMB (Figure S5; TraCE-21ka, MPI, and IPSL), simulated a stronger magnitude and a wider swath of cold based ice conditions across the Adirondacks (Figure S4; TraCE-21ka, MPI, and IPSL). Such conditions are similar to what we observe across the thick ice divides of modern ice sheets (i.e., Greenland Ice Sheet; MacGregor et al., 2022), where in the absence of heat generation due to frictional heating, vertical advection of cold surface climate dictates the basal thermal state (Lai and Anders, 2021). We also find that frozen bed conditions existed across other areas of high bedrock relief during the LGM (Figure 3) where frozen bed patches are simulated above warm-based ice at lower elevations. Across both the White Mountains and Mount Katahdin (Figure 5&6), upstream ice flow slowed as it encountered resistance from the underlying high bedrock relief. Only where both ice velocity and thickness are relatively low, and bedrock elevation is high, is the presence of frozen bed conditions simulated. Where ice thickness and driving stresses increase downstream of these bedrock features, ice velocities increase and warm-based conditions are simulated.

Regional geologic interpretations support that a thermal boundary between cold-based (low erosive) ice and warm-based (erosive ice) existed at ~1200 m across areas of high bedrock relief in the NE USA (Halsted et al., 2022). Terrestrial cosmogenic nuclide (TCN) surface exposure ages from various peaks across the NE USA exhibit signs of nuclide inheritance which is interpreted to reflect inefficient erosion by previous ice coverage. TCN ages from the peaks of Mount Katahdin (ME), Mount Washington (NH), and Little Haystack Mountain (NH), exhibit signs of nuclide inheritance suggestive of a thermal boundary below those locations (Bierman et al., 2015). While warm-based ice indicators (e.g., roche moutonnée and lodgement till) are found on Mt. Washington between 1680 m and 1820 m, a lack of age control on those features means the timing of erosion relative to the LGM is inconclusive. Across Mount Katahdin and the White Mountains, we



simulate a thermal boundary of 1318 m and 1530 m respectively, which is in reasonable agreement
with the geologic assessment.
Across the Adirondack Mountains, frozen bed conditions are simulated both in areas of
high bedrock relief and lower elevation sites that reside under a simulated regional ice divide,
making the interpretation more complicated. While the region wide thermal boundary is simulated
to be at 882 m, if we only consider the High Peaks Area where geologic data of ice thinning exists
(Barth et al., 2019), the thermal boundary resides at 1180 m. For high elevation sites, Barth et al.
(2019) present an alternative hypothesis that the high-elevation TCN ages suggest early ice-sheet
thinning ~20 ka in lieu of reflecting nuclide inheritance and that the regional thermal boundary
likely existed above 1560 m. Nevertheless, our simulations confirm that this thermal boundary was
not spatially constant, and instead varied geographically (Bierman et al., 2015; Corbett et al., 2018;
Koester et al., 2021).
While our experiments offer a spatially high-resolution reconstruction of the LGM basal
thermal conditions of the LIS across the NE USA that agree well with independent assessments
from geologic reconstructions, our methodology assumes that the LIS is in equilibrium with the
LGM surface climate, which is likely not a true reflection of LGM conditions. It is noted that
while Tarasov and Peltier (2007) simulate the LIS thermal state across the last glacial cycle and
find that maximum areal coverage of frozen bed conditions occurred during the LGM, since the
LIS experienced a transiently evolving climate prior to the LGM as the surface climate cooled, our
simulations may reflect a colder thermal state than may have been experienced. Additionally, the
spatially varying basal friction coefficient is constant and not thermodynamically coupled in our
simulations. While this coupling has recently been shown to have greatest influence in areas of
ice streaming across the LIS with attendant feedbacks on ice surface lowering and ultimately ice
thickness, a thermodynamically coupled friction may promote colder basal conditions through
feedbacks between ice flow, ice temperature, and basal friction (Moreno et al., 2023). Although
we cannot fully address how these limitations and assumptions affect our results, since our
simulations agree well with geologic interpretations that polythermal conditions and ultimately a
regime of differential erosion existed across the NE USA, we attain a level of confidence that we
are indeed simulating thermal conditions that generally reflected LGM conditions. However,
future work should evaluate how these thermal conditions may have changed in response to
deglacial LIS change as our contemporary assessment of geomorphic and geologic indicators
likely integrates the full glacial and deglacial history.
Because of the difficulties in conducting dipstick studies aimed at constraining vertical
thinning histories, our regional and local scale modeling framework may prove helpful for making
more informed choices on sample site selection in places where model simulations suggest warm-
based, and ultimately erosive ice conditions, such is currently being done for fieldwork in the
Adirondack Mountains (Barker et al., 2024). Additionally, since the results presented here support
broader geologic interpretations that polythermal ice conditions likely existed across the NE USA
(Halsted et al., 2023), such output may be useful in geomorphological interpretations of differential
erosion and relief generation as well as transport processes of glacial erratics from lower elevation,
warm-based areas (Bierman et al., 2015). Lastly, such a frame work shows promise in applications
to other regions of the LIS where geologic and geomorphic indicators suggest the existence of
sharp thermal contacts and erosional history, such as across portions of Arctic Canada and Baffin
Island (Briner et al., 2006; 2014).



### *5. Conclusions*

In this study, we use a numerical ice sheet model to simulate at high spatial resolution, steadystate LGM basal thermal conditions for the LIS across the NE USA and at 3 specific locations characterized by high bedrock relief. LGM climate boundary conditions are used from a small ensemble of climate model simulations, each with a varying degree of LGM cooling and precipitation change relative to preindustrial climate. Our results illustrate that during the LGM, the LIS across the NE USA was mainly warm-based and ultimately erosive, yet exhibited polythermal ice conditions, as simulations reveal that cold-based ice existed across this region in areas of high bedrock elevation and slow ice flow (i.e., ice divides). At local scales, we find that within the Adirondack Mountains, a regional ice divide is simulated during the LGM, characterized by low ice velocities (<15 m/yr) and a wide swath of cold-based ice that spans a large elevational range. Across the White Mountains and Mount Katahdin, ice velocities are generally higher, with cold-based ice conditions being simulated only amongst the highest elevation peaks. Where existing models (Marshall and Clark, 2002; Tarasov and Peltier, 2007; Gregoire et al., 2012) lack sufficient resolution to capture these features, for the first time we simulate a complex thermal regime that may have existed across this region reflective of the highly variable topography of the region.

The results presented here support the conclusions from a large dataset of TCN surface exposure ages that relate nuclide inheritance across high relief areas of the NE USA to the presence of cold-based and low erosive conditions sometime during the LGM and last deglaciation (Halsted et al., 2023). These studies largely support that a thermal boundary of ~1200 m in elevation separated cold-based ice at higher elevations and warm-based ice at lower elevations. While our simulations support this conclusion, they also illustrate that this thermal boundary was not spatially consistent and instead varied geographically. Additionally, the results here are supportive of lower latitude polythermal ice conditions existing across the LIS during the LGM (Bierman et al., 2015; Colgan et al., 2002). The existence of polythermal ice conditions across this region has implications with respect to glacial geomorphology, as the erosive character of the ice sheet is closely tied to the basal thermal regime. Since dipstick studies (Bierman et al., 2015; Koester et al., 2017; Barth et al., 2019; Corbett et al., 2019; Koester et al., 2020; Halsted et al., 2023) have the potential to provide critical constraints on paleo ice sheet thinning, yet interpretations can be hindered by the existence of cold-based ice (i.e. low erosion and thus nuclide inheritance), studies like this may aid in future study site selection (e.g. Briner et al., 2022; Barker et al., 2024) as this downscaling procedure can be applied to specific sites of interest. Additionally, this downscaling approach may be useful for geomorphological assessments in other areas of the LIS where polythermal conditions may have existed (Davis et al., 1999; Davis et al., 2006; Briner et al., 2014; Staiger et al., 2005), by providing another metric to evaluate landscape evolution.

**Code and data availability**
The simulations performed for this paper made use of the open-source Ice-Sheet and Sea-level System Model (ISSM) and are publicly available at https://issm.jpl.nasa. gov/ (Larour et al., 2012). Model output described in this study can be found at https://doi.org/10.5281/zenodo.12665418 (Cuzzone et al., 2024). This includes the simulated output of LGM ice velocity (x and y components as well), ice thickness, and the simulated thermal agreement for cold and warm-based ice across the NE USA, the Adirondack Mountains, the White Mountains, and Mount Katahdin.



**Author contributions.** JC, AB, and KB conceived the study. JC conducted the model setup and conducted the experiments with input from MM. JC analyzed model output with help from AB, KB, and MM. JC and AB wrote the manuscript with input from KB and MM.

**Competing interests:** The contact author has declared that none of the authors has any competing interests.

**Financial support:** This work was supported by a grant from the National Science Foundation, Division of Earth Science (EAR; grant no. 2133699).

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
