# Peer review of "Ice sheet model simulations reveal polythermal ice conditions existed across the NE USA during the Last Glacial Maximum"

_EGUsphere, 2024_

## Referee Comment (RC2)

[referee-annotated manuscript omitted]

---

## Author Comment (AC1)

**Review of "Ice sheet model simulations reveal polythermal ice conditions existed across the NE USA during the Last Glacial Maximum" by Cuzzone et al.**

I apologise for the late return of this review, and any inconvenience it may have caused the authors or editor.

Cuzzone et al. here present simulations of the Laurentide Ice Sheet, using small regions of high horizontal resolution to explore the basal thermal regime in areas of variable topography. Simulations at a continental scale require such coarse resolution that the thermal effect of mountainous regions may be excluded. In this paper, extremely high-resolution simulations demonstrate a polythermal bed and sharp thermal gradients in regions across the eastern Laurentide Ice Sheet.

Overall, the authors present some impressive simulations with interesting results. The work is well-described and presented, with some minor corrections to the figures suggested below. The research is clearly a good fit for the journal, and the work is worthy of publication.

We would like to thank Niall Gandy for the review and encouragement of our results. We are much appreciative for your time taken to help improve our manuscript. We have taken into account the reviewers' comments to improve some of our figure clarity and have modified some text (particularly discussion) to address some of the points raised below. Thanks again for your help.

**Major points**

1. A mini-ensemble is presented, with simulations using boundary conditions from 5 different climate models, but I am left wondering if we are confident that the variation between climate model output is certainly the first-order uncertainty to be explored. Perhaps this is a reasonable assumption, but I think this should be discussed in the text. If varying other parameters (such as elements of the PDD scheme of the basal sliding law) would be produce larger variation, plots like figure 3 may provide a false confidence in the certainty of the results.

The PDD factors can/do influence the overall surface mass balance. In general, less melt and higher rates of snowfall in the ice sheet interior will lead to an increase in vertical advection. This would cause the vertical distribution of ice temperature to trend colder, and may result in increased potential for frozen bed conditions. However, the PDD factors do not have a large influence across this domain during the LGM. During the LGM surface temperatures were below freezing across most months. Only near the ice sheet margins did summertime temperature exceed freezing. Therefore, there is little to no snow melt across the interior of the ice sheet at the LGM across this domain, and the effect of the pdd factors on surface melt and accumulation are limited to marginal areas where summertime melt occurs during the LGM. From a climate perspective, the input climatology of temperature and precipitation influence the basal thermal state as these forcings influence the surface temperature and accumulation. We show this in Figure S5 and describe in text (lines 445-453), in those simulations with lower LGM temperature, higher accumulation, and ultimately higher SMB, the basal conditions are colder (more frozen bed).

To clarify this better, we adjusted the text in lines 474-477 to read: *"Those experiments with colder LGM boundary conditions and higher accumulation simulated higher SMB (Figure S5; TraCE-21ka, MPI, and IPSL) and therefore resulted in a stronger magnitude and a wider swath of cold based ice conditions across the Adirondacks (Figure S4; TraCE-21ka, MPI, and IPSL)."*

This is not to say the PDD factors do not have an impact. It is just that this impact is limited during the LGM when sfc. Temperatures are below freezing. It is possible that the PDD factors may have an increasing influence during the deglaciation, when sfc. Temperature rises and melt across interior locations ensues. Therefore we have added text on lines 487-509 to discuss this:

*"We note that while the impact of degree day factors is unexplored in this work, we expect this to have a limited influence on the simulated thermal characteristics of the LIS across the NE USA. During the LGM, monthly surface temperatures remain below freezing across the ice sheet interior, with exception of the Southern margin (Figure S5). Therefore, surface melt is limited across the ice sheet interior, and instead the prescribed surface temperature and accumulation are the primary factors driving the simulated differences in the thermal conditions (Figure S5). Nevertheless, for simulations across the last deglaciation, the choice of degree day factors can have a large influence on the simulated ice history and ultimately the transient evolution of ice temperature as deglacial surface temperature and melt rise (Matero et al., 2020)."*

With respect to the basal sliding law, we performed an additional simulation following our experimental setup, but instead used the Assimilated Weertman-Coulomb Friction Law (otherwise known as Schoof Law: Schoof et al., 2005). We find that the influence on the simulated steadystate basal temperatures are small (on the order of <5% difference) when compared to the basal temperatures simulated using the Budd friction law. Because of this, we do not think the choice of friction law would strongly influence the conclusions reached in our manuscript. However, the choice of friction law may become increasingly important during transient simulations across the last deglaciation. This question has not been addressed in current literature, but is something we will pursue in future work when performing transient simulations across the last deglaciation.

[Figure]

A subpoint is that it would be useful to have a sense of the computational expense of these simulations, both to help the planning of future studies and perhaps to further justify the limited size of the ensemble.

In general our models are around 200,000 elements. We note that we use a higher order stressbalance approximation. Simulations takes ~4-12 days for a LGM relaxation to steadystate (NE Model), although with iterative solvers (which we are experimenting) this can be cut in half. Smaller models are between 200,000-300,000 elements. Relaxation is shorter however as model reaches convergence faster.

2.  A broader justification of the work would be useful in both the Introduction and Conclusion. Understanding the thermal effect of small-scale topography may be important for understanding the past evolution of LGM ice sheets, and future evolution of Greenland and Antarctica. But that is my interpretation, and it would be better to state your justification more explicitly. There is mention of models providing geologic constraints across a wide area, but I'm clear on what this would be useful for (because of my own ignorance, I'm sure!).

Thanks for this question. The ability to resolve basal topography can have large impacts on the simulated ice sheet history due to its impact on grounding line stability and ultimately ice retreat/advance as well as attendant feedbacks on the ice thermal regime. While this is an important consideration, we do not aim to frame our results in this manner. Perhaps future work could address this as we move towards transient simulations across this domain.

Ultimately, we hope our work is first useful for geologists/geomorphologists to help infer basal conditions where geologic data is limited.  We hope that this conundrum was clear in our Introduction (Paragraph #2) and Discussion/Conclusions.  The issue at hand is that some geologic data suggests the presence of cold-based ice conditions, and ultimately incomplete erosion.  This becomes apparent when looking at cosmogenic surface exposure ages that have inheritance issues, which is particularly abundant in high elevation sites.  Dipstick studies that aim to constrain vertical thinning histories across the NE USA suffer from this issue in some regions, making interpretations of vertical thinning difficult.   Our results present a tool (from model outputs) that may inform where geologists sample along vertical transects, by choosing peaks that may have been (at least simulated to have been) warm based ice and erosive.  While this model informed sampling technique has yet to be put to the test (co-authors Barth and Barker on this paper will use these outputs in field sampling summer 2025), it may become a tool for future sampling campaigns not only across the NE USA but other areas where polythermal conditions existed.  While we did not want to overstate this as a tool, we did provide text in the Discussion and Conclusion highlighting this.  If the reviewer does indeed want further clarification, we would be happy to add more, but feel our current justifications are adequate.

3.  There are a few points where you compare your results to geologic evidence or other empirical evidence (some specific examples are highlighted below). A more robust quantitative comparison may be insightful, or at least a discussion of why this might not be possible.

We agree that a more robust comparison between the existing geologic/isotopic data would be beneficial. However, for the purposes of this manuscript that may be difficult. Any geochronologic data (e.g., TCN exposure ages) from the high peaks demonstrate nuclide inheritance and provide little benefit in constraining a time component. Additionally, as our model simulations are solely concentrated on the LGM, any geochronologic data younger than the LGM, that indicate warm-based ice conditions, are not yet relevant and would require a transient simulation of the deglaciation. Therefore, geologic/isotopic data are primarily considered spatially as indicators of basal thermal conditions with correlations drawn against modeling results. As this approach is not statistically robust, we have softened some of the language (e.g., "…simulations *broadly* agree with geologic interpretations…") and hope that addresses the reviewers' concerns.

**Minor points**

Figure 1: Could the model nesting be incorporated into this figure, rather than a separate figure in the SI?

After trying to satisfy this request, we would like to keep these figures separate as we could not find a suitable fix that was visually appropriate.

Line 167: Adjustment for clarity? "…independent of the thermal state but *this* has been explored…"

Requested change has been made.

Line 167: I'm not sure I'm following the final sentence of this paragraph. The differences between simulations with and without thermal-friction coupling are measurable but small? Could this be quantified?

Here we are referring to discussion in Moreno et al., 2023, in that the thermal-friction coupling tends to have little impact across the NE USA domain in those experiments.

Line 201: Is "downscale" the accepted word here? I (rightly or wrongly) interpret that as a statistical process.

We believe the appropriate usage is dynamical downscaling. We have modified the text.

Line 207: It seems odd that these locations are partly informed by state boundaries that didn't exist during (and have no impact on) glaciation. Would a purely geomorphic apolitical process have identified the same location?

Because geomorphic studies are at state level they are informed by local entities research (e.g. state geologists, etc), and often refer to these data (citations with our manuscript) at state level. Making our results most amenable to those fields we decided to keep as is but have adjusted the text in line 206 to differentiate that our choice relies on the highest elevation in each state:

*"The three locations were chosen based on their geomorphic qualities, as each represents three of the highest peaks in the NE USA, and has pre-existing geologic data on glaciation (e.g., cosmogenic surface exposure ages; Bierman et al., 2015; Barth et al., 2019)."*

Line 228: And is this climate data also used as the surface temperature flux to calculate the ice sheet thermal state?

Yes, the surface temperature is prescribed directly at the ice surface. We have updated the text ~ line 254:

*"The thermal regime is assumed to be in steady-state with the LGM climate, and we perform a thermomechanical steady state calculation with a fixed ice sheet geometry and apply surface air temperatures directly at the ice surface until the ice sheet velocities are consistent with ice temperature (i.e., convergence) following Seroussi et al. (2013)."*

Line 228: Given the focus of the paper I think it is entirely reasonable PDD values are prescribed and unexplored, as long is it produces a SMB field that serves the simulations. An additional justification may be useful for some readers, though.

Paleoclimate remains one of the largest uncertainties in paleo ice sheet model boundary conditions. While the PDD factors can influence the SMB and ultimately the ice sheet geometry (e.g. thickness, extent) and ultimately ice velocity and temperature, we feel that by using a range

of plausible climate states we capture with some fidelity a range of simulated ice conditions. As described in the response above (Point #1), during the LGM, surface temperatures across the ice sheet interior remain below freezing during all months.  Therefore, surface melt is limited across the ice interior, and the influence of the PDD factors on the ice temperature is likewise limited. It has an effect closer to the ice margin where melt is simulated (see Figure S5 for SMB maps), but this is not critical to our conclusions regarding the thermal characteristics of the LIS across this region, particularly at high (bedrock) elevation sites across the ice interior.

To clarify further, we have added some text to our discussion Line 461:

*"We note that while the impact of degree day factors is unexplored in this work, we expect this to have a limited influence on the simulated thermal characteristics of the LIS across the NE USA. During the LGM, monthly surface temperatures remain below freezing across the ice sheet interior, with exception of the Southern margin (Figure S5).  Therefore, surface melt is limited across the ice sheet interior, and instead the prescribed surface temperature and accumulation are the primary factors driving the simulated differences in the thermal conditions (Figure S5). Nevertheless, for simulations across the last deglaciation, the choice of degree day factors can have a large influence on the simulated ice history and ultimately the transient evolution of ice temperature as deglacial surface temperature and melt rise (Matero et al., 2020)."*

Section 2.2: Could this process be summarised in a diagram or table in the SI? I found myself drawing in my notebook to help keep track of the process.

Would something like this be of use to make things more clear?  If so, we could put this into the supplemental and reference it as a quickguide.

1.) 2D LIS Model:  Spinup with constant LGM climate to arrive at equilibrium ice geometry and flow.
2.) 3D LIS Model:  Extrude the 2D model to 3D
    a.  Conduct a thermomechanical steady state computation to initialize model ice temperature and rheology.
    b.  Perform an additional relaxation with constant LGM climate to allow model ice temperature and rheology to adjust to ice geometry and flow.
3.) NE USA Model:  Construct a 3D model of the NE USA.
    a.  Interpolate ice geometry, temperature, rheology, and velocity from LIS 3D model onto NE USA model mesh.
    b.  Perform a thermomechanical steady state computation to downscale the coarser (LIS model) initial temperature and rheology inputs onto the higher resolution NE USA domain.
    c.  Perform a relaxation with constant LGM climate to allow ice geometry, velocity, temperature, and rheology to adjust to higher resolution model mesh and reach equilibrium.
4.) Local Models: Construct 3D model of local domains (Adirondacks, Mt. Washington, and Mount Katahdin)
    a.  Interpolate ice geometry, temperature, rheology, and velocity from NE USA 3D model onto local model meshes.

b. Perform a thermomechanical steady state computation to downscale the coarser (NE USA model) initial temperature and rheology inputs onto the higher resolution local model domains.

c. Perform a relaxation with constant LGM climate to allow ice geometry, velocity, temperature, and rheology to adjust to higher resolution model mesh and reach equilibrium.

Figure 2: Could colour-blind friendly ramps by used on ice velocity figures? The use of white stippling is nice.

We are more than happy to adjust this per the reviewer's request, however, in making this colormap initially we consulted with the BYU colorblind image tester and found the colormap to be colorblind friendly. https://bioapps.byu.edu/colorblind_image_tester.

If needed we would be happy to change, but feel the current colormap is satisfactory.

[Figure]

Figure 4: This figure contains a lot of useful information but it takes some work to fully appreciate! To improve readability, you could include a location panel to save readers bouncing between figures? The position and scale of various axis and legend labels could be refined to increase readability. The figure is a bit small, but panel B doesn't add much to panel E. If you

removed B the remaining panels could be enlarged. A scale bar would be useful alongside the graticule.

Figure 5: As above.

Figure 6: And again.

We agree with the reviewer that the figures here could be modified to improve readability. Thanks for the help here.

Following your recommendations, we opted to remove the 'panel B' and replace that with 'panel E' from our original submission. There are now 4 panels which have allowed us to enlarge the overall figure size – which we hope improves readability. A scale bar and inset map showing the location of the subdomain has been added to panel A.

Line 417: "Our results agree with geologic interpretations". Is it possible to provide a more robust statistical or graphical comparison between your results and geological interpretations?

To address this, we refer to our answer to point 3 above.

Line 424 onwards: It would be interesting to see a comparison of the simulated velocity field at different scales? If these regions are important, I suppose it would have a measurable impact on the flux across the domain?

Our apologies, but we are unclear what/where the reviewer is referring to with the different scales and regions? We think the comparisons between Figures 2&3 and the local model figures (4,5,6) illustrate these differences adequately. Here we are referring to the resolution of topographic features in coarse resolution models, and the inability to capture the thermal gradients that existed across the high gradients in terrain. The boundary conditions for the higher resolution simulations come from our coarser regional models, which still are pretty high resolution. They capture the general ice flow (direction, speed; Figure 2 & 3) across the NE USA well. These boundaries do not disagreement in cross boundary fluxes (i.e. coarse model simulating flow into or out of domain where high res. model would simulate the opposite), so therefore we do not expect the downscaling to be impacted by the boundary conditions.

---

## Author Comment (AC2)

This is a really interesting manuscript that models the thermal status of the now-vanished Laurentide Ice Sheet (LIS) in the northeastern United States and Canada. Model results provide insight into the thermal conditions at the base of the ice sheet which can be compared to both geologic observations of glacial erosion and isotopic/geochemical measurements of proxies such as cosmogenic nuclides.

I am not a modeler so have focused most of my comments on the less technical side of the text although reading the modelling section did bring up several ideas that I have made comments about. Given my lack of expertise in the modelling domain I posed these as questions that hopefully the authors can address because many of the readers of this work may have similar questions.

I attach a marked up PDF from my reading and summarize below what I consider to be the most important points to address in revision of the manuscript. I believe addressing these suggestions will make the manuscript more readable to a broader audience and have tried where possible to offer specific guidance to the authors. If the authors have questions about my review, I encourage them to contact me. Paul Bierman

We would like to thank Paul Bierman for his thorough review of our manuscript, and appreciate your time taken to help improve our manuscript. We have gone through your comments (in PDF) and updated the editorial comments raised and recommended. Because the review was done as PDF comments, we have taken those comments, written them down here with our answers below. We first address the larger points as follows:

1. The manuscript should be more inclusive in its referencing of prior work. Some earlier papers are not yet cited and there are claims made that are without citations. I've made suggestions on the PDF and the authors can find more citations to add.

Thank you for taking the time to point out additional citations. We have noted in our response to comments (in the attached PDF) where these changes have been made following your guidance.

2. The abstract could be much stronger and more informative. There are ambiguous statements and words that leave the reader uncertain. I suggest a significant and careful edit on the abstract after the paper is revised. See specific suggestions on PDF.

Thank you for the help and recommendations to clarify the text in the abstract. You will see in the response to your comments (attached PDF) and the tracked changes, we have made edits to adjust the clarity of the abstract.

3. While adding references, I suggest the authors look more deeply into the geomorphic and soils literature of New England. There are papers relating to preserved saprolite that are relevant as well as to striations at and near the peaks of most ranges suggesting that at some point the ice was indeed warm-based. Similarly there is striation mapping near coast. Adding a more inclusive discussion of the landscape will provide deeper context for their results. It will also illuminate the importance of transience something that must be key for the LIS but is not yet discussed (and should be). I allude to this in a question on the ms about model spin up.

We have updated the text to include a more detailed description of the regional geomorphic features and their role as warm- vs. cold-based ice indicators.

"as requested

*"Glacial striations and roche moutonnée found below the summits of the Presidential range in New Hampshire and Mt. Katahdin in Maine indicate the presence of warm-based ice locally (Goldthwait, 1970; Davis, 1978; 1989; Thompson et al., 2002). Similar evidence with the addition of weakly developed soils on the summits of the Presidential Range indicate cover by erosive, warm-based ice (Goldthwait, 1970)."*

*"Furthermore, a lack of precise age control for many of these features leaves the question of temporal variability unanswered and whether or not the thermal regime was consistent throughout glaciation and deglaciation."*

Transience is something we acknowledge in the current manuscript.  You will see in track changes and our response to your comments that we have further acknowledged this assumption.

4.  The ms currently moves between spatial changes (plan view) and elevation changes but not in a coherent and predictable fashion. The story will be easier to follow and will benefit from more clarity and an organization structure that clearly and separately considers the elevation effect vs latitude effect on basal thermal regime.

We appreciate this comment and the suggestion to reorganize the manuscript to distinguish between spatial and elevational changes. However, we opted to leave the manuscript organized as is since the focus of our work was to evaluate elevational changes solely, consider those for various regions in the NE USA, and not the latitudinal effect on the thermal regime. We do not include any discussion of changes in the basal thermal regime nearer the margins instead focusing on mountainous regions that would have experienced similar ice-coverage histories (i.e., deglaciated around the same time).

5.  From an ease of reading perspective, some paragraphs are really long and contain multiple foci. Breaking into shorter paragraphs with topic sentences will make the paper easier for the reader to understand. Similarly, there are places (I have tried to call them out in the text) where wording is inexact and thus could confuse readers. I have done my best in the PDF to suggest places it could be improved for clarity. There are sections of the discussion that read like methods and likely should be moved to the methods section or removed.

    Yes, thank you for the editorial suggestions. Following your guidance, we have inserted page breaks where necessary and moved/removed some text in the discussion.  Please see our response to your comments (attached PDF) for point-by-point responses.

6.  The long time of model relaxation seems to violate known ages and geologic history of ice in New England. This should be addressed explicitly. See extended note on PDF.

    We think that this point raised has been addressed in our discussion, and the reviewer made a comment to address this point further up in the methods.  Rather than move this text (we think it is proper to talk about this assumption in the discussion section), we also added text at the end of the methods section to illustrate the acknowledgement of this shortcoming in our model setup.

7.  The precision of elevation boundaries to the nearest meter seems to overstate the actually precision of the modelling and the topographic data. Suggest rounding to at least the 10s if not the hundreds of meters given all of the uncertainties.

    Yes, thanks for the recommendation – we agree.  We have updated the precision.

8. I agree strongly with the first reviewer that more direct comparison between the model and geologic/isotopic data sets would be very useful and strengthen the paper.

   We agree that a more robust comparison between the existing geologic/isotopic data would be beneficial. However, for the purposes of this manuscript that may be difficult. Any geochronologic data (e.g., TCN exposure ages) from the high peaks demonstrate nuclide inheritance and provide little benefit in constraining a time component. Additionally, as our model simulations are solely concentrated on the LGM, any geochronologic data younger than the LGM, that indicate warm-based ice conditions, are not yet relevant and would require a transient simulation of the deglaciation. Therefore, geologic/isotopic data are primarily considered spatially as indicators of basal thermal conditions with correlations drawn against modeling results. As this approach is not statistically robust, we have softened some of the language (e.g., "…simulations *broadly* agree with geologic interpretations…") and hope that addresses the reviewers' concerns.

Response to Reviewer comments in PDF ( we have taken the pdf comments and placed them here):

- Line 12 Abstract:  The abstract needs a careful edit. The language and geographic descriptions are ambiguous and will confuse readers not familiar with the specific prior literature and cosmogenic nuclides.
  - Changes have been made as requested, and we have shortened/edited the abstract.
- Line 16:  not clear of what? presume you mean cosmogenic nuclides. this should be made clear to the reader.
  - Change has been made as requested.
- Line 20: which? this is ambiguous
  - Removed 'These'
- Line 21:  I would think of that as Long Island, NJ and such but data also suggest summits up ice. Suggest broadening.
  - Broadened by stating 'NE USA'
- Line 25:  what region?not well defined. High summits? southernmost extent?
  - Changed to 'NE USA'
- Line 29:  I think you mean interpreted from...people interpret data don't.
  - Change has been made as requested.
- Line 30: higher than what? seems like jargon
  - This is a common term in ice sheet modeling, but we removed this to avoid confusion. It is not necessary to our statement here.
- Line 37: data are plural
  - Change has been made as requested.
- Line 42:  finer and higher need comparatives - presumably finer and higher than older models?
  - We have updated the text as recommended.
- Line 51:  and geographically!  that's a big sea level change. I'd make a nod to that.
  - We have updated the text as recommended.
- Line 60: There are several new Halsted papers that would strengthen this list - especially the JQS paper comparing chronometers.

Halsted, C. T, Bierman, P. R., Shakun, J. D., Davis, P. T., and Corbett, L. B., (2023) A critical re-analysis of constraints on the timing and rate of Laurentide Ice Sheet recession in the northeastern United States, Journal of Quaternary Science. 110.1002/jqs.3563

Also, there is the Corbet NJ terminal moraine area paper with dates to diversify this list and the second Balco paper (CT coast) and a more obscure Drebber paper. I would make this an inclusive rather than selective list of cosmo papers in NE, there are not that many.

Corbett, L. B., Bierman, P. R., Larsen, P., Stone, B. D. and Caffee, M. W. (2017) Cosmogenic nuclide age estimate for Laurentide Ice Sheet recession from the terminal moraine, New Jersey, USA, and constraints on Latest Pleistocene ice sheet history. Quaternary Research. v. 87(3), p. 482-498. doi.org/10.1017/qua.2017.11

> Drebber, J., Halsted, C., Corbett, L., Bierman, P., and Caffee, M. (2023) In-situ cosmogenic 10Be dating of Laurentide Ice Sheet retreat from central New England, USA, Geosciences 2023, 13(7), 213 doi.org/10.3390/geosciences13070213
> - Thank you for the help and recommendations. We have updated the list.

- Line 71: We replaced 'difficult' with 'uncertain'
- Line 74: I think that you meant the GSAB paper here for dipsticks?

Halsted,C., Bierman,P., Shakun, J., Davis, P. T., Corbett, L., Caffee, M., Hodgdon, T., and Licciardi, J. (2022) Rapid southeastern Laurentide Ice Sheet thinning during the last deglaciation revealed by elevation profiles of in-situ cosmogenic 10Be, GSA Bulletin. doi.org/10.1130/B36463.1
> - Yes, thank you for the correction. We have updated this reference.

- Line 81: erosion depth though controls cosmo inheritance it's a combination of time and rate which needs to be disentangled.
  - Does the reviewer suggest changing this text? Erosional depth can be related to erosional patterns, so we would keep the text as is for now – but happy to change if needed.
- Line 93: Break Paragraph, new idea
  - Change has been made as requested.
- Line 103: Break Paragraph, new idea
  - Change has been made as requested.
- Line 103: need to define this for reader...and from a pedantic view, I'd argue that the archive is rock and the proxy that informs us is primarily cosmogenic nuclides but there's also a much older literature on weathering and saprolite preserved in NE which must indicate lack of deep erosion! Worth digging into that.
  - Defined "geologic archives" earlier in the paragraph.

    *"Geologic archives such as glacial erratics, striations, and soil development, as well as isotopic evidence from terrestrial cosmogenic nuclide (TCN) surface exposure ages, can be interpreted to reflect subglacial thermal regimes (i.e., cold- or warm-based ice) in the past."*
- Line 114: above it's archives - best to be consistent
  - We have changed to 'archives' to be consistent.
- Line 115: I see no need to support - why not evaluate! support seems like a preconceived notion.
  - We have changed 'support' to 'evaluate'
- Line 120: Removed, 'comprising of the states listed in'
- Line 198: please define for reader how you define low and high with a specific metric.
  - We have updated the text to refer to 'gradients' in bedrock topography.
- Line 200: per above comment, best to be consistent...is this the archive of previous pages? geomorphic data is not cosmogenic.
  - Have edited to read 'geologic archives'

- Line 207: but isotope data are not geologic...perhaps geochemical? geologic would be moraine mapping, rock types...
    - We have changed the text to 'geochemical'
- Line 274: I am not a modeler and may be misunderstanding this approach but the LIS did not occupy New England for 20,000 years. Existing data suggest that it crossed over the US border about 30,000 years ago and that the margin was largely north of the border by 13,000. thus is was not in steady state thermally. It may be that I am off base here (in which case please explain why this "relaxation" is valid or that it's impossible to model transience (which is how I read this) but at least there needs be a discussion of what this deviation from the geologic reality might mean for results - some kind of sensitivity test would be great! and really illustrative for the reader.
    - We discuss the limitations in conducting a 'steadystate' simulation of LGM conditions in the Discussion section: see lines 518 and beyond. While it is true that the ice sheet at the LGM may not have been in true steady state, its characteristics likely reflect a more complex history. As the ice advanced southward, ice temperature would have been the result of the accumulating new ice (which would reflect the transient surface climate conditions) and advection of ice from upstream. Therefore, it is difficult to assess how this may influence results. However, our methodology is similar to how ice sheets are initialized currently, and how we assess the basal conditions of the present-day ice sheets (Greenland; see MacGregor et al., 2016 (referenced within)). These reconstructions are taken with the caveat that they may not capture the thermal history exactly as the surface conditions were transiently evolving. Nevertheless, we included text outlining this caveat in our discussion. *We note the reviewer commented positively on this in our discussion, and suggested to move this text to end of this section.

    - Ultimately, we left the text in our discussion and instead have also added some more text at the end of the methods section to address this:

      "This methodology has been applied to study the basal temperature of present-day ice sheets (Seroussi et al., 2013; MacGregor et al., 2016; 2020), and although it assumes the ice sheet is in equilibrium with climate, we acknowledge that the thermal conditions of the LIS during the LGM likely reflect transiently evolving ice geometry and climatic conditions experienced during the growth and advance to the LGM maximum. "

- Line 318: suggest paragraph break here, new idea.
    - We have corrected this as recommended.
- Line 403: really should cite Marsella's very large data set from Baffin that discusses polythermal ice and the extension of that data set by Corbett. See:

  Corbett, L. B., Bierman, P. R. and Davis, P.T. (2016) Glacial history and landscape evolution of southern Cumberland Peninsula, Baffin Island, Canada, constrained by cosmogenic 10Be and 26Al. Geological Society of America Bulletin. v. 128(7-8), p. 1173-1192. doi.org/10.1130/B31402.1

  Marsella, K. A., Bierman, P. R., Davis, P. T. and Caffee, M. W. (2000) Cosmogenic 10Be and 26Al ages for the last glacial maximum, eastern Baffin Island, Arctic Canada. Geological Society of America Bulletin. v. 112(8), p. 1296-1312. doi.org/10.1130/0016-7606(2000)112<1296:CBAAAF>2.0.CO;2
    - Thank you for these references. They have been added.

- Line 413-416: this reads more like methods than discussion

o   Agreed. We changed the sentence to reference previous mention in the Methods.

*"Our downscaled, local simulations broadly agree with geologic interpretations (Davis, 1989; Bierman et al., 2015; Corbett et al., 2018; Halsted et al., 2023) that suggest cold-based ice existed across areas of high relief in the NE USA and the existence of polythermal ice for the region during the LGM."*

- Line 418:  and Corbett's Mansfield paper...

  Corbett, L. B., Bierman, P. R., Wright,S., Shakun, J., Davis, P.T., Goehring, B., Halsted, C., Koester, A., Caffee, M., and Zimmerman, S. (2018) Analysis of multiple cosmogenic nuclides constrains Laurentide Ice Sheet history and process on Mt. Mansfield, Vermont's highest peak, Quaternary Science Reviews. doi.org/10.1016/j.quascirev.2018.12.014
     o   Thank you.  We have added that reference.

- Line 424: I think that there are Gosse data from here and you should look at in review paper by Cavnar and the cites in it for more suggestions of inheritance. As well the work of LeBlanc. See:
     Cavnar, P.M., Bierman, P.R., Shakun, J.D., Corbett, L. B., LeBlanc, D., Galford, G.L., and Caffee, M. (in review, 7.2024) In situ Cosmogenic 10/Be and 26Al in deglacial sediment reveals interglacial exposure, burial, and limited erosion under the Quebec-Labrador Ice Dome. Geochronology. doi.org/10.5194/egusphere-2024-2233
     LeBlanc, D. E., Shakun, J. D., Corbett, L. B., Bierman, P. R., Caffee, M. W., and Hidy, A. (2023) Laurentide Ice Sheet persistence during Pleistocene interglacials, Geology. doi.org/10.1130/G50820.1
     o   Thanks for the references, but this paper we cited has a study area which matches closely with the simulated frozen bed - that is why we chose to cite this paper in particular.

- Line 425:  data plural, "are"
     o   Change has been made.
- Line 432: Removed "when looking at" per reviewers request.
- Line 432-437:  this reads more like methods than discussion
     o   Have removed the sentence:  "Our experiments relied on a small ensemble of simulated surface climate at the LGM from climate model experiments, each simulating a varying magnitude of LGM cooling and precipitation change."
- Line 437:  you don't observe this - others report. reword.
     o   Change has been made as recommended.  "others report"
- Line 448-457:  this is all background that would be more appropriate as a set up in the intro than here in the discussion.
     o   Deleted most of this section and made specific reference to the individual peaks in the updated introduction. Updated sentence to:

*"Regional geologic interpretations of isotopic data suggest that a thermal boundary between cold-based (low erosive) ice and warm-based (erosive ice) existed at ~1200 m across areas of high bedrock relief in the NE USA (Halsted et al., 2022). Yet, undated geomorphic indicators of warm-based ice on Mount Washington are found at 1680 m and 1820 m."*

- Line 458:  is this level of precision really achievable? Suggest at least rounding to the 10s of meters.
     o   We have rounded to the nearest 10s and updated for all other references to the precision in the text.

- Line 465: this is confusing...it's more a statement of fact (perhaps better in intro) than a decision of which might be right?
    - This sentence was left as is as we believe it acts as a direct comparison to the available geologic data regarding location of the thermal boundary as is done for the other local simulations (e.g., Mt. Katahdin and Mt. Washington). We believe this sentence acknowledges a discrepancy with one interpretation of the data, yet, still supports a variable thermal boundary for the NE USA which is demonstrated in our regional simulations.
- Line 471: I think this should be "agrees"
    - Change has been made.
- Line 473: if it agrees well, then you should cite these reconstructions and elaborate - are they geologic? isotopic?
    - Removed reference to the geologic reconstructions in this paragraph. The goal of this paragraph is to acknowledge that climate was not in a steady state - unlike what is simulated in our model. The connection to geologic reconstructions is not necessary here and is discussed earlier in the paper.

- Line 480: Ok, good to see this. I might move it up to near where I queried assumptions or even into methods.

    - Yes, we hope this helps answer some of the questions posed previously by the reviewer. Instead of moving this up we prefer to keep as is since it is a discussion point of some caveats in our experimental setup. We have instead opted to add additional text to the end of the methods section. See our response to your comment raised above in the methods section (Line 274).
- Line 490: they are not difficult to conduct...but the data are difficult to interpret. that's different. please reword for accuracy.
    - Good point. We have updated the text as recommended.
- Line 497: first mention of erratics vs bedrock - probably should be part of the intro
    - Reference to glacial erratics now made in the intro with reference to studies by De Laski, 1872; Tarr, 1900; Antevs, 1932; Davis, 1989.
- Line 498: really should cite earlier work including Marsella et al that suggested this.
    - Yes, thanks for the recommendation - done.
- Line 518: no need to be self-congratulatory – reader will get this! It's a cool paper on its own.
    - Change made as requested.
- Line 523: yes, but there are others and they all should be mentioned! especially the earlier ones by other authors.
    - We have updated with more references.
- Line 527: but the references cited are not lower latitude (or not substantially) and one is in the midwest on a lobe underlain by far less topography. It's also continental climate not adjacent to the ocean - the conclusion is not the place to discuss this but the discussion is!
    - Upon re-reading we agree, it seems out of place. Have removed.